# Particle-based Online Bayesian Sampling

## Abstract

Online learning has gained increasing interest due to its capability of tracking real-world streaming data. Although it has been widely studied in the setting of frequentist statistics, few works have considered online learning with the Bayesian sampling problem. In this paper, we study an Online Particle-based Variational Inference (OPVI) algorithm that updates a set of particles to gradually approximate the Bayesian posterior. To reduce the gradient error caused by the use of stochastic approximation, we include a sublinear increasing batch-size method to reduce the variance. To track the performance of the OPVI algorithm concerning a sequence of dynamically changing target posterior, we provide the first theoretical analysis for the dynamic regret from the perspective of Wasserstein gradient flow. Experimental results on the Bayesian Neural Network show that the proposed algorithm achieves up to 20% improvement than naively applying existing Bayesian sampling methods in the online setting.

## 1 Introduction

Online learning is an indispensable paradigm for problems in the real world, as a machine learning system is often expected to adapt to newly arrived data and respond in real-time. The key challenge in this setting is that the model cannot be updated with all data in history each time, which grows linearly and would make the system unsustainable. There are quite a few online optimization methods developed over the decades that address the challenge by only taking the last arrived batch of data for each update and by using a shrinking step size to control the increase of error. They have been successfully applied to a wide range of tasks like online ranking, network scheduling and portfolio selection (Yu et al., 2017; Pang et al., 2022).

Online optimization methods can directly be applied to update models that are fully specified by a certain value of its parameters. Beyond such models, there is another class of models known as Bayesian models that treat the parameters as random variables, thus giving an output also as a random variable (often the expectation is taken as the final output on par with the conventional case). The stochasticity enables Bayesian models to provide diverse outputs, characterize prediction uncertainty, and be more robust to adversarial attacks (Hernández-Lobato & Adams, 2015; Li & Gal, 2017; Zhang et al., 2019). However, traditional Bayesian models, particularly previous Bayesian sampling methods, typically focus on the offline case, where revisiting past data is required.

In contrast, online statistics can be computed using only the recently arrived streaming data points. The online Bayesian sampling issue is significant, driven by numerous application scenarios where data is collected sequentially, necessitating the real-time update of the uncertainty-aware learning model. For instance, certain types of data, such as Magnetic Resonance Imaging (MRI), are challenging to acquire, making a continuous update scheme essential for problems like Bayesian matrix/tensor completion Zhang & Hawkins (2018). Other scenarios include Bayesian on-device learning in uncertain environments with streaming sensor data Servia-Rodriguez et al. (2021), where it is impractical to obtain the entire dataset at once. Nevertheless, the learning procedure of Bayesian models is different from conventional models, which poses a challenge in directly applying online optimization methods in an online setting. This is because a Bayesian model is characterized by the distribution of its parameters but not a single value, and the learning task, a.k.a. Bayesian inference, is to approximate the posterior distribution of the parameters given received data. A tractable solution is Variational Inference (VI) (Jordan et al., 1999; Blundell et al., 2015), which approaches

the posterior using a parameterized approximating distribution, which enables optimization methods again (Hoffman et al., 2010a; Broderick et al., 2013a; Foti et al., 2014). However, the accuracy is restricted by the expressiveness of the approximating distribution which is not systematically improvable.

A more accurate method is Monte Carlo which aims to draw samples from the posterior. As the posterior is only known with an unnormalized density function, direct sampling is intractable, and Markov chain Monte Carlo (MCMC) is employed. While it makes sampling tractable, it comes with the issue of sample efficiency due to the correlation among the samples. Recently, a new class of Bayesian inference methods is developed, known as particle-based variational inference (ParVI) (Liu & Wang, 2016; Chen et al., 2018; Liu et al., 2019; Zhu et al., 2020; Zhang et al., 2020; Korba et al., 2021; Liu & Zhu, 2022). They try to approximate the posterior using a set of particles (i.e., samples) of a given size, which are iteratively updated to minimize the difference between the particle distribution from the posterior. The accuracy of the method can be systematically improved with more particles, and due to the limited number of particles, sample efficiency is enforced so as to minimize the difference. The theoretical nature, under the offline case, has been widely studied over the Wasserstein space, where the Wasserstein gradient flow has emerged as a promising approach to solve optimization problems over the space of probability distributions. While ParVI methods have been successfully applied to the full-batch and mini-batch settings, to our knowledge there is no online version of ParVI.

In this work, we develop an Online Particle-based Variational Inference (OPVI) method to meet this desideratum and also provide an analysis on its regret bound which can achieve a sublinear order in the number of iterations. The method and analysis are inspired by the distribution optimization view of ParVI on the Wasserstein space, under which we could leverage techniques and theory of conventional online optimization methods. To do this, we first extend existing Maximum a Posterior (MAP) methods to handle the prior term better and then extend the results to the Wasserstein space as an online sampling method. Generally, our contribution can be summarized as follows:

- We propose the new online sampling methods by employing an increasing batch-size scheme and better prior-term settings that improve upon naively applying existing ParVI methods in an online setting.

- We give the theoretical analysis for the OPVI method, which is the first theoretical analysis for ParVI from the perspective of dynamic regret by leveraging the Riemannian structure of the Wasserstein space.

- We give a detailed analysis of the gradient error led by stochastic gradient descent methods, which is important for the type of ParVI methods with injected diffusion noise.

- We conduct empirical tests for the OPVI method. The results suggest much better posterior approximation and classification accuracy than naive online ParVI methods and other MCMC methods.

## 2  Related Work

Since (Cesa-Bianchi & Lugosi, 2006) study the online properties of VI, there are a couple of works showing online VI gives good performance in practice cases (Hoffman et al., 2010b; 2013; Broderick et al., 2013b). Furthermore, researchers in (Chérief-Abdellatif et al., 2019) derive the theoretical results for the generalization properties of the Online VI algorithm. Even though online VI is well studied, few papers pay attention to the problem of online MCMC. A potential alternative to online MCMC could be Sequential Monte Carlo (SMC) methods (Chopin, 2002; Kantas et al., 2009; Christensen et al., 2012), which combine importance sampling with Monte Carlo schemes to track the changing distribution. However, SMC does not have the Markov chain properties of MCMC, such as convergence to a stationary distribution, thus, impossible to obtain theoretical guarantees of asymptotic convergence to the true posterior distribution. In general, no previous work considers an online MCMC method from the perspective of optimization methods, which provides a possible way to find the theory behind the convergence.

Our method employs a gradient descent-based optimization strategy to update particles toward the target posterior. However, the target posterior is dynamically changing with streaming data arriving in the system, which makes the optimal solutions change. To solve this problem, we consider a performance metric called dynamic regret in our analysis. To achieve a sublinear dynamic regret, researchers propose different constraints on the sequence of loss functions, like the gradient variation (Rakhlin & Sridharan, 2013), and path variation (Yang et al., 2016). However, even though this dynamic problem is essential to be considered in the analysis of Bayesian inference algorithms, no previous papers considered this. As a result, existing theoretical guarantees regarding the online VI (e.g. (Chérief-Abdellatif et al., 2019)) may be insufficient under the dynamic changing online environment.

The stochastic gradient descent algorithm is widely used as an incremental gradient algorithm that offers inexpensive iterations by approximating the gradient with a mini-batch of observations. Through the past decade, it has been used in a wide variety of problems with different variations, like network optimization (Pang et al., 2022; Zhou et al., 2022) reinforcement learning (Liu et al., 2021b;a), federated learning (Sun & Wei, 2022) and recommendation system (Yang et al., 2020). However, this method, at the same time, incurs gradient error when approximating the gradient. In most of the novel sampling methods, we normally obtain diverse solutions by injecting diffusion noises (e.g. Langevin Dynamic (LD) (Neal et al., 2011), Stochastic Gradient Langevin Dynamics (SGLD) (Welling & Teh, 2011), which makes this type of algorithm sensitive to the noise. This instability makes reducing the stochastic gradient error important.

To reduce the gradient error, researchers studied multiple variance reduction methods, like using adaptive learning rates and increasing batch size. In the previous work, an adaptive learning rate was used to adapt the optimization to the most informative features with Adagrad (Ward et al., 2019) and estimate the momentum for Adam(Kingma & Ba, 2014). Compared with the adaptive methods, the increasing batch size methods have greater parallelism and shorter training times (Smith et al., 2017) and are also studied in offline and online cases (Friedlander & Schmidt, 2012; Zhou et al., 2018), which shows great importance in achieving applicable convergence rate and sublinear regret bound.

## 3 The online Maximum a Posterior on Euclidean Space $\mathcal{W}$

In this section, we first introduce an online MAP algorithm on Euclidean decision space $\mathcal{W}$ with gradient descent method, which helps the reader to understand our OPVI sampling method on Wasserstein space. Here, we give some prior knowledge about the online MAP problem and the dynamic regret metric. Then, we give a detailed policy using an online stochastic gradient descent algorithm to solve the online MAP problem and a detailed theoretical analysis based on the dynamic regret metric.

### 3.1 Preliminaries

For an online MAP algorithm run with time slots $t \in [1, T]$, let $\mathcal{W} \in \mathcal{R}^d$ denote a convex set, set $w_t \in \mathcal{W}$ be some parameter of interest and $\mathcal{N}_T = \{\mathcal{B}_1, \cdots, \mathcal{B}_t, \cdots, \mathcal{B}_T\}$ be the set of i.i.d. observations, where $\mathcal{B}_t = \{d_k, d_{N_t}\}$ is a batch of $N_t$ data samples arrived at time $t$. In a typical problem of MAP, we aim to maximize a target posterior $p(w) := p_0(w) \prod_{k=1}^{N_T} p(d_k \mid w)$, where we usually take logarithm on both sides to simplify the computation as $\log p(w) = \log p_0(w) + \sum_k \log p(d_k|w)$.

Different from the offline MAP, we set a $\eta_t = \frac{6}{\pi^2 t^2}$ adaptive weight for the prior in our online setting, which divides the whole prior for each update with $\sum_{t=1}^{T} \eta_t = 1$ when $T \to \infty$. Then, the goal of the online MAP problem on $\mathcal{W}$ is to find parameter $w_t$ that maximizes the cumulative of a linear combination of minus likelihood and partial prior, which can be given as:

$$\max_{w_t \in \mathcal{W}} \sum_{t=1}^{T} \Big( \sum_{d_k \in \mathcal{B}_t} \log p(d_k \mid w_t) + \eta_t \log p_0(w_t) \Big), \tag{1}$$

To simplify the notation, we use $c_t^k(w_t) := -\log p(d_k \mid w)$ to denote the log-likelihood with data $d_k$ and $c_0(w_t) := -\log p_0(w_t)$ to denote the log-prior, where $c$ is called the cost function in the literature of optimization and we take minus logarithm since we want to make sure the cost function to be positive all the

time. We denote $c_t(w_t) = \sum_{k=1}^{N_T} c_t^k(w_t)$ as the true likelihood considering all data in the dataset. Then, we can formulate the goal eq. (1) to be an optimization problem with $c_t + \eta_t c_0$ as the objective function and follow the goal of:

$$\min_{w_t \in \mathcal{W}} \sum_{t=1}^{T} c_t(w_t) + \eta_t c_0(w_t)$$

As we have mentioned in Section 2, the target posterior is dynamically changing with the new observations, we are interested in using dynamic regret as the performance metric for our problem, which is defined as the difference between the total cost incurred at each time slot and a sequence of optimal solutions $\{w_t^*\}$ in hindsight, i.e.,

$$R(T) = \sum_{t=1}^{T} c_t(w_t) + \eta_t c_0(w_t) - c_t(w_t^*) - \eta_t c_0(w_t^*). \tag{2}$$

## 3.2 Dynamic Algorithm for Online Maximum a Posterior

It is well known that the online gradient descent algorithm can be used to solve online optimization problems (Zinkevich, 2003; Besbes et al., 2015; Yang et al., 2022). Here, we give an online stochastic gradient descent algorithm for the online MAP problem in the following updating policy:

$$\mathbf{w}_t = \begin{cases} \mathbf{w}_1 \in \mathcal{W} & t = 1 \\ \Pi_{\mathcal{W}}[w_{t-1} - \alpha v_t(w_{t-1})] & t > 1 \end{cases}, \tag{3}$$

$$\text{where } v_t(w_{t-1}) = \nabla c_{t-1}(w_{t-1}) + \eta_t \nabla c_0(w_{t-1})$$

where $\Pi_{\mathcal{W}}$ is the projection back to the convex set $\mathcal{W}$. projection, need an additional inequality in proof Next, we first introduce some widely used assumptions required for the theoretical analysis.

**Assumption 1.** (Bounded Convex Set) For a convex decision set $\mathcal{W}$ and any two decisions $w_1, w_2 \in \mathcal{W}$, we have $d(w_1, w_2) \leq R$.

**Assumption 2.** (Convexity and Lipschitz smooth) The function $c_t + \eta_t c_0$ is convex and Lipschitz smooth, so its derivatives are Lipschitz continuous with constant $L$ with a constant $L$, i.e., for two real $w_1, w_2 \in \mathcal{W}$, we have:

$$\|\nabla c_t(w_1) + \eta_t \nabla c_0(w_1) - \nabla c_t(w_2) - \eta_t \nabla c_0(w_2)\| \leq L\|w_1 - w_2\| \quad t \in [1, T].$$

**Assumption 3.** (Vanishing gradient) We assume the optimal solutions $w_t^*$ lie in the interior of the convex set $\mathcal{W}$, where we assume there exists $w_t^*$ such that $\nabla c_t(w_t^*) + \eta_t \nabla c_0(w_t^*) = 0$

We give a sublinear regret upper bound in the next subsection, which means $\|w_t - w_t^*\|$ is decreasing and the parameter of interest $w_t$ can converge to the dynamic changing optimal solutions $w_t^*$ when $T$ is large enough. That indicates we can obtain a promising MAP result with the policy in eq. (3).

## 3.3 Theoretical Analysis for Online MAP

In this section, we begin with the proof of the online MAP algorithm following the policy in eq. (3) over the Euclidean space $\mathcal{W}$. As we mentioned in Section 2, it is impossible to achieve a sublinear regret bound for any sequence of cost functions. To solve this problem, we consider a path variation budget $V_T$ for the sequence of optimal solutions $\{w_t^*\}$, which bound the cumulative path length of the optimal solutions as $V_T := \sum_{t=1}^{T} \|w_t^* - w_{t-1}^*\|$. The result is summarized in Theorem 4, which gives the sublinear bound for the dynamic regret $\mathcal{R}(T)$ when we set a sublinear path variation $V_T$.

**Theorem 4.** *Under the Assumption 1 - 3, given a sequence of optimal solutions $\{w_t^*\}$, variational budget $V_T$, following the updating policy in eq. (3) on Euclidean Space $\mathcal{W} \in \mathcal{R}^n$, we have the dynamic regret:*

$$\mathbb{E}[\mathcal{R}(T)] \leq \frac{R^2}{\xi} + \frac{RV_T}{\xi},$$

*where $\xi = 2\alpha - 2L\alpha^2$*

**Proof.** Detail of the proof can be found in Appendix A. □

The theorem presents the regret bound for the online MAP problem within the Euclidean space, where the bound primarily depends on the variational budget $V_T$. Setting $V_T$ as a sublinearly increasing term allows the regret bound to increase sublinearly with respect to $T$. Next, we will introduce the proposed OPVI method and extend the proof of online MAP over the Wasserstein space.

## 4 Online Particle-based Variational Inference on Wasserstein Space $\mathcal{P}_2(\mathcal{W})$

In this section, we propose the OPVI algorithm on $\mathcal{P}_2(\mathcal{W})$, which formulate the online MAP problem in Section 3 as an online sampling method from the perspective of Wasserstein gradient flow. To begin with, we first introduce some preliminary knowledge about the 2-Wasserstein Space $\mathcal{P}_2(\mathcal{W})$, as well as its Riemannian structure and the gradient flow on it. Then, we give a brief introduction to a well-known ParVI method, called SVGD (Liu & Wang, 2016) and take it as an example to illustrate how to simulate a ParVI problem as a gradient flow on $\mathcal{P}_2(\mathcal{W})$. Based on this idea, we give the theoretical analysis for OPVI as a distribution optimization flow on $\mathcal{P}_2(\mathcal{W})$ to show a sublinear dynamic regret. For convenience, we only consider Wasserstein Space supported on the Euclidean space $\mathcal{W}$ in our analysis.

Here, we first clarify the notation used in this section. We use $\mathcal{C}_c^\infty$ as a set of compactly supported $R^D$−valued functions on $\mathcal{W}$ and use $C_c^\infty$ to denote the scalar-valued functions in $\mathcal{C}_c^\infty$. Except for the Euclidean space $\mathcal{W}$ and Wasserstein space $\mathcal{P}_2(\mathcal{W})$ we just mentioned, we consider two other types of space in this paper, the Hilbert space $\mathcal{L}_q^2$ and the vector-valued Reproducing Kernel Hilbert Space (RKHS) $\mathcal{H}^D$ of a kernel $K$. The Hilbert space $\mathcal{L}_q^2$, is a space of $\mathbf{R}^D$-valued functions $\left\{ u : \mathbb{R}^D \to \mathbb{R}^D \mid \int \|u(w)\|_2^2 \, \mathrm{d}q < \infty \right\}$ with inner product $\langle u, v \rangle_{\mathcal{L}_q^2} := \int u(w) \cdot v(w) \mathrm{d}q$. The RKHS $\mathcal{H}$ is a kernel version of the Hilbert space $\mathcal{L}_q^2$, which is the closure of linear span $\{f : f(w) = \sum_{i=1}^m a_i K(w, w_i), a_i \in \mathbb{R}, m \in \mathbb{N}, w_i \in \mathcal{W}\}$ equipped with inner products $\langle f, g \rangle_{\mathcal{L}_q^2} = \sum_{ij} a_i b_j K(w_i, w_j)$ for $g(w) = \sum_i b_i K(w, w_i)$.

### 4.1 The Wasserstein Space $\mathcal{P}_2(\mathcal{W})$, its Riemannian Structure and the Gradient Flow

Generally, the Wasserstein space is a metric space equipped with Wasserstein distance $d(\cdot, \cdot)$. Set $P(\mathcal{W})$ as the space of probability measures on the Euclidean support space $\mathcal{W}$. The 2-Wasserstein space on $\mathcal{W}$ can be defined as $\mathcal{P}_2(\mathcal{W}) := \{\mu \in P(\mathcal{W}) : \int_{\mathcal{W}} \|w\|^2 d\mu(w) < \infty\}$. Since the Riemannian structure of Wasserstein space is discovered (Otto, 2001; Benamou & Brenier, 2000), several interesting quantities have been defined, like the gradient and the inner product on it.

To define the gradient of a smooth curve $(q_t)_t$ on $\mathcal{P}_2(\mathcal{W})$, we can set a time-dependent vector field $v_t(w)$ on $\mathcal{W}$, such that for a.e. $t \in \mathbb{R}, \partial_t q_t + \nabla \cdot (v_t q_t) = 0$ and $v_t \in \overline{\{\nabla\varphi : \varphi \in C_c^\infty\}}^{\mathcal{L}_{q_t}^2}$, where the overline means closure (Villani, 2009). Note that the vector field $v_t$ here is the so-called tangent vector of the curve $(q_t)_t$ at $q_t$ and the closure is denoted as tangent space $T_{q_t}\mathcal{P}_2$ at $q_t$, whose elements are the tangent vectors for the curves passing through the point $q_t$. The relation between $T_{q_t}\mathcal{P}_2$, $v_t$ and $\mathcal{P}_2(\mathcal{W})$ can be found in Fig. 1. The inner product in the tangent space $T_{q_t}\mathcal{P}_2$ is defined on $\mathcal{L}_q^2$, which defines the Riemannian structure on $\mathcal{P}_2(\mathcal{W})$ and is consistent with the Wasserstein distance due to the Benamou-Brenier formula (Benamou & Brenier, 2000).

An important role of the vector field representation is that we can approximate the change of distribution $q_t$ within a distribution curve $(q_t)_t$. For a single update in each time slot, we can set $(\mathrm{id} + \varepsilon v_t)_\# q_t$ as a first-order approximation of the updated distribution $q_{t+1}$ in the next time slot (Ambrosio et al., 2005). Therefore, for a set of particles $\{x_t^{(i)}\}_i$ that obey distribution $q_t$ at time $t$, we can update these particles with a stepsize of $\varepsilon$ as $\{x_t^{(i)} + \varepsilon v_t(x_t^{(i)})\}_i$, to approximate distribution $q_{t+1}$ in time $t+1$, when $\varepsilon$ is small. We show this approximation as a red arrow in Fig. 1. In the task of Bayesian inference, our goal is to minimize the KL-divergence between a current estimated distribution $q_t$ and the target posterior $p$ as $KL_p(q_t) := \int_{\mathcal{W}} \log(q_t|p)dq_t$, which has the tangent vector for its gradient flow $(q_t)_t$ as a vector field of:

$$v_t = -\nabla_{q_t}\mathrm{KL}_p(q_t) = \nabla \log p - \nabla \log q_t,$$

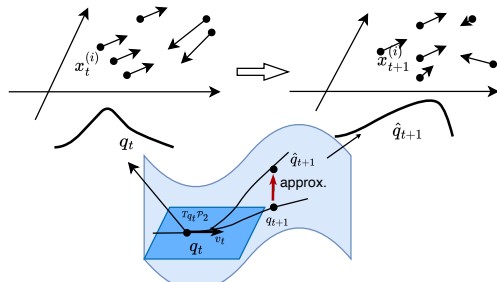

Figure 1: Illustration for the updating of $q_t$ over the gradient flow $(q_t)_t$ on $\mathcal{P}_2(\mathcal{W})$, and the relationship between the update of particles $\{x_t^{(i)}\}$ over $\mathcal{H}$ and the update of distribution $q_t$ over $\mathcal{P}_2(\mathcal{W})$.

### 4.2  Particle-based Variational Inference Methods

In this section, we first use SVGD as an example to illustrate the ParVI methods. Then we show how to simulate SVGD as the gradient flow on Wasserstein space $\mathcal{P}_2(\mathcal{W})$, which can help the analysis of OPVI in the following subsection. For SVGD, let $\{x_t^{(i)}\}_{i=1}^n$ be a set of particles that obey an empirical measure of distribution $q_t$. We initialize $q_t$ as some simple distribution $q_0$, then use a vector field $v$ to update these particles toward the target posterior $p$: $x_{t+1}^{(i)} = x_t^{(i)} + \varepsilon v(x_t^{(i)})$, where $v$ should be chosen to maximize the decreasing of the KL-divergence $- \frac{\mathrm{d}}{\mathrm{d}\varepsilon}\mathrm{KL}_p\left((\mathrm{id} + \varepsilon v)_{\#}q\right)\big|_{\varepsilon=0}$. In SVGD, the vector field is chosen to be optimized over RKHS $\mathcal{H}$ with a closed-form solution:

$$v_{\mathcal{H}}^{\mathrm{SVGD}}(\cdot) := \nabla \log p(x)k(x,\cdot) + \nabla k(x,\cdot) \tag{4}$$

Note that the updating of SVGD particles is actually an approximation of the $\mathcal{P}_2(\mathcal{W})$ gradient flow by taking $\mathcal{H}$ as its tangent space instead of $\mathcal{L}_{q_t}^2$, since the function in $\mathcal{H}$ is roughly a kernel smoothed function in $\mathcal{L}_{q_t}^2$ (Liu & Zhang, 2019). Thus, the vector field $v_{\mathcal{H}}^{\mathrm{SVGD}}$ in eq. (4) can be used to approximate the vector field $v_{\mathcal{L}_{q_t}^2}^{\mathrm{SVGD}}$ in $\mathcal{L}_{q_t}^2$ on $P_2(\mathcal{W})$ (Liu et al., 2019, Theorem 2), where the solution gives:

$$v_{\mathcal{H}}^{\mathrm{SVGD}} = \max \arg \max_{v \in \mathcal{H}, \|v\|_{\mathcal{H}}=1} \langle v_{\mathcal{L}_{q_t}^2}^{\mathrm{SVGD}}, v \rangle_{\mathcal{L}_{q_t}^2} \tag{5}$$

That enables us to use $v_{\mathcal{L}_{q_t}^2}^{\mathrm{SVGD}}$ to approximate the vector field $v_{\mathcal{H}}^{\mathrm{SVGD}}$ on $P_2(\mathcal{W})$ in the following analysis, which like doing a projection from $\mathcal{H}$ to $\mathcal{L}_{q_t}^2$.

### 4.3  Online Particle-based Variational Inference on $\mathcal{P}_2(\mathcal{W})$

In this section, we aim to develop an online sampling method on $\mathcal{P}_2(\mathcal{W})$ and proposed the OPVI algorithm. We first illustrate the policy of OPVI over RKHS $\mathcal{H}$. Then, we interpret the OPVI as the gradient flow on $\mathcal{P}_2(\mathcal{W})$ and conduct the theoretical analysis by transferring the proof in 3 from Euclidean space $\mathcal{W}$ to Wasserstein space $\mathcal{P}_2(\mathcal{W})$. Note that we use $v_t^{\mathrm{OPVI}\text{-}\mathcal{H}}$ as the vector field on RKHS $\mathcal{H}$ and $v_t^{\mathrm{OPVI}\text{-}\mathcal{L}^2}$ as the vector field on $\mathcal{L}_{q_t}^2$.

We begin with reviewing the minus KL-divergence in an offline setting, which is given as:

$$-KL_p(q_t) = \mathbb{E}_{q_t}[\log p] - \mathbb{E}_{q_t}[\log q_t] = \sum_{k=1}^{N_T} \mathbb{E}_{q_t}[\log p(d_k|\cdot)] + \mathbb{E}_{q_t}[\log p_0] - \mathbb{E}_{q_t}[\log q_t],$$

where $N_T$ is the number of data samples in the dataset. Following a similar idea as the online MAP algorithm we introduced in section 3, we set a $\eta_t = \frac{6}{\pi^2 t^2}$ adaptive weight for the prior in our online setting and using mini-batch with batch size $B_t$ to approximate the likelihood.

However, the approximation leads to a gradient error between the true gradient $\nabla \log p$ and the estimated gradient $\nabla \hat{\log} p$, which gives $e_t := \nabla \hat{\log} p - \nabla \log p$, which can be calculated by:

$$e_t = \frac{1}{B_t} \sum_{k \in \mathcal{B}_t} \nabla \log p(d_k|\cdot) - \nabla \log p, \tag{6}$$

where $B_t = |\mathcal{B}_t|$ is the batch size. Also, we define the error leads by the wrongly added prior term as $h_t$, which can be defined as $h_t := \hat{\nabla} \log p - \nabla \log p$. For convenience in proof, we define the $\epsilon_t$ to represent the combination noise of $e_t$ and $h_t$, which is given as $\epsilon_t := e_t + \eta_t h_t$. Note that the gradient error $e_t$ can be deterministic or stochastic, depending on the way we set up the mini-batch. In this paper, we choose to select samples for mini-batch $\mathcal{B}_t$ arbitrarily from $\mathcal{N}_T$, which makes the gradient error stochastic. We introduce an error bound $E_T$ to measure the cumulative gradient error lead by the stochastic gradient approximation over $t \in [1, T]$, which is given by:

$$E_T := \sum_{t=1}^{T} \epsilon_t \tag{7}$$

We will show a sublinear increasing batch size is enough to keep $E_T$ growing sublinear, which enables the online MAP algorithm to enjoy a sublinear dynamic regret.

Thus, we give an online stochastic version of minus KL-divergence between $q_t$ and the dynamic changing posterior $p_t$ as:

$$-\text{O-KL}_{p_t}(q_t) = \sum_{k=1}^{B_t} \mathbb{E}_{q_t}[\log p(d_k|\cdot)] + \eta_t \mathbb{E}_{q_t}[\log p_0] - \mathbb{E}_{q_t}[\log q_t] \tag{8}$$

Similar to SVGD, we first draw a set of particles $\{x_0^{(i)}\}_{i=1}^n$ that obey some simple initial distribution $q_0$. Then, we update these particles with a gradient descent updating scheme with step size $\alpha$:

$$x_{t+1}^{(i)} = x_t^{(i)} + \alpha v_t^{\text{OPVI-}\mathcal{H}},$$

where $v_t^{\text{OPVI-}\mathcal{H}}$ is the vector field on $\mathcal{H}$ that maximizes the decrease of online stochastic KL-divergence $-\frac{d}{d\alpha}\text{O-KL}_{p_t}((id + \alpha v_t)_{\#q})|_{\alpha=0}$ to give a closed-form solution:

$$v_t^{\text{OPVI-}\mathcal{H}}(\cdot) = \mathbb{E}_{q(x)}[K(x,\cdot)\nabla \sum_{k=1}^{B_t} \log p(d_k|x) + \eta_t K(x,\cdot)p_0(x) + \nabla K(x,\cdot)],$$

where $K(x,x')$ is satisfied by commonly used kernels like the exponential kernel $K(x,x') = \exp(-\frac{1}{h}\|x - x'\|_2^2)$ and the general workflow of the OPVI algorithm is summarized in Alg. 1.

## 4.4 Proof of Dynamic Regret Bound under $\mathcal{P}_2(\mathcal{W})$

To begin with, we first formulate the updating rule in Alg. 1 as a Wasserstein gradient flow. Here, we ignore the kernel smooth used in the implementation of the algorithm by approximating the vector field $v_t^{\text{OPVI-}\mathcal{H}}$ on RKHS $\mathcal{H}$ with the vector field $v_t^{\text{OPVI-}\mathcal{L}^2}$ on Hilbert space $\mathcal{L}^2$. To simplify the proof, we denote $c_t^k(q_t) = -\mathbb{E}_{q_t}[\log p(d_k|\cdot)]$ and $c_t^0(q_t) = -\eta_t \mathbb{E}_{q_t}[\log p_0] + \mathbb{E}_{q_t}[\log q_t]$ in eq. (8) and follow eq. (6) to represent the stochastic approximation as the sum of the true gradient and a gradient error $e_t$, which gives:

$$v_t^{\text{OPVI-}\mathcal{L}^2}(q_t) = -(c_t(q_t) + e_t + c_t^0(q_t))$$

Then, the updating of the particles can be formulated as an optimal transport for distribution $q_t$ over $\mathcal{P}_2(\mathcal{W})$ as:

$$q_{t+1} = \text{Exp}_{q_t}(-\alpha(c_t(q_t) + e_t + c_t^0(q_t))) \tag{9}$$

Before we give the proof for the regret bound, we first re-assume some assumption under the $\mathcal{P}_2(\mathcal{W})$.

---

**Algorithm 1** Online Particle-based Variational Inference

---

Initialize particles $\{x_0^{(i)}\}_{i=1}^N$
**for** $t = 1, \cdots, T$ **do**

$$x_{t+1}^{(i)} = x_t^{(i)} + \alpha v_t^{\text{OPVI-}\mathcal{H}}(x_t^{(i)})$$

where:

$$v_t^{\text{OPVI-}\mathcal{H}}(x_t^{(i)}) = \mathbb{E}_{q(x)}[K(x, x_t^{(i)})\nabla \sum_{k=1}^{B_t} \log p(d_k|x) + \eta_t K(x, x_t^{(i)})\nabla \log p_0(x) + \nabla K(x, x_t^{(i)})]$$

**end for**

---

**Assumption 5.** (Bounded geodesically-convex set) Assume $\mathcal{K}$ to be a g-convex set on some Wasserstein space $\mathcal{P}_2(\mathcal{W})$ supported on $\mathcal{W}$. From Theorem 2 of (Gibbs & Su, 2002), we can establish a bound for the maximum Wasserstein distance in a bounded support space with $\dim(\mathcal{W}) < R$. Then $\forall q_1, q_2 \in \mathcal{P}_2(\mathcal{W})$, we have:

$$d_{\mathcal{K}}(q_1, q_2) \leq 1 + R$$

which bound the geodescially convex set $\mathcal{K}$.

**Assumption 6.** (Geodesically-L-Lipschitz (g-L-Lipschitz)). Similar to the definition over $\mathcal{W}$, we assume $c_t(q_1) + c_t^0(q_1)$ to be a g-convex function and has a geodesically L-Lipschitz continuous gradient on $\mathcal{P}_2\mathcal{W}$ if there exists a constant $L > 0$ that:

$$|\nabla c_t(q_1) + \eta_t \nabla c_t^0(q_1) - \nabla c_t(q_2) - \eta_t \nabla c_t^0(q_1)| \leq L \cdot d(q_1, q_2), \quad \forall q_1, q_2 \in \mathcal{P}_2(\mathcal{W}),$$

where $d(a, b)$ should be some Wasserstein distance.

Compared with the proof on $\mathcal{W}$, the key difference is the way to obtain Lemma 9. Instead of updating a set of parameters of interest over $\mathcal{W}$, we update the distribution $q_t$ by optimal transport over $\mathcal{P}_2(\mathcal{W})$.

**Lemma 7.** *Suppose that $\mathcal{P}_2(\mathcal{W})$ is a Wasserstein space supported on Euclidean space $\mathcal{W}$ with the sectional curvature lower bounded by $-\kappa(\kappa > 0)$. Under Assumption 3, 5, 6, for any $q_t \in \mathcal{K}$, following the updating rule in eq. (9), we have:*

$$\mathbb{E}[d(q_{t+1}, q_t^*)] \leq \mathbb{E}[d(q_t, q_t^*)] - \frac{\Phi}{R}\mathbb{E}[(c_t(q_t) + \eta_t c_t^0(q_t) - c_t(q_t^*) - \eta_t c_t^0(q_t^*))] + \sqrt{2\alpha\epsilon_t^2\zeta(\kappa, R) + 2\alpha\epsilon_t R},$$

*where $\Phi = 2\alpha - 3L\alpha^2\zeta(\kappa, R)$.*

***Proof.*** We start from a fact proved in Lemma 6 of (Zhang & Sra, 2016), which gives an inequality for a geodesic triangle with curvature bounded by $\kappa$, where the length of sides for the triangle is $a$, $b$, $c$ and $A$ is the angle between sides $b$ and $c$, then:

$$a^2 \leq \frac{\sqrt{|\kappa|}c}{\tanh(\sqrt{|\kappa|}c)}b^2 + c^2 - 2bc\cos(A),$$

In our work, we map our problem on a triangle, where the vertices of this triangle is set to be three status of the decisions in our problem, the current step decision $q_t$, the next step decision $q_{t+1}$ and the optimal solution in current step $q_t^*$. Denote $d(a, b)$ to be the Wasserstein distance between two distribution $a$ and $b$ over $\mathcal{P}_2(\mathcal{W})$. As a result, the three sides of the triangle should be $a = d(q_{t+1}, q_t^*)$, $b = d(q_t, q_{t+1})$ and $c = d(q_t, q_t^*)$. Base on the updating rule, we have $d(q_t, q_{t+1}) = \alpha\|\nabla c_t(q_t) + e_t + \nabla c_t^0(q_t)\|$ when $\alpha$ is small enough and $d(q_t, q_{t+1})d(q_t, q_t^*)\cos(\angle q_{t+1}q_t q_t^*) = \langle -\alpha(\nabla c_t(q_t) + e_t + \nabla c_t^0(q_t)), \text{Exp}_{q_t}^{-1}(q_t^*)\rangle$.

The detailed proof can be found in Appendix B $\qquad\square$

Using Lemma 7 and the definition of dynamic regret in eq. (2), we give the dynamic regret bound on $\mathcal{P}_2(\mathcal{W})$ in the following Theorem.

**Theorem 8.** *(Regret Bound over $\mathcal{P}_2(\mathcal{W})$) Under the Assumption 3, 5, 6, given a sequence of optimal solutions $\{q_t^*\}$, define the variational budget $V_T := \sum_{t=1}^{T} d(q_t^*, q_{t+1}^*)$ and the error bound $E_T$. Following the updating rule in eq. (9), we have the dynamic regret bound:*

$$\mathcal{R}_{\mathcal{P}_2(\mathcal{W})} \leq \mathcal{O}(\max(1, E_T, V_T)) \tag{10}$$

**Proof.** The proof start from Lemma 7 with using the triangle inequality, which gives:

$$\mathbb{E}[d(q_{t+1}, q_t^*)] \leq \mathbb{E}[d(q_{t+1}, q_t^*)] + \mathbb{E}[d(q_{t+1}^*, q_t^*)]$$

$$\leq \mathbb{E}[d(q_t, q_t^*)] - \frac{\Phi}{R}\mathbb{E}[(c_t(q_t) + \eta_t c_t^0(q_t) - c_t(q_t^*) - \eta_t c_t^0(q_t^*))] + \sqrt{2\alpha\epsilon_t^2 \zeta(\kappa, R) + 2\alpha\epsilon_t R} + \mathbb{E}[d(q_{t+1}^*, q_t^*)]$$

Rearrange the inequity, taking summation from $t \in [1, T]$, evolving the definition of the regret, we have:

$$\mathbb{E}[\mathcal{R}_{\mathcal{P}(\mathcal{W})}(T)] \leq \frac{R}{\Phi}\sum_{t=1}^{T}\mathbb{E}[d(q_t, q_t^*) - d(q_{t+1}, q_t^*)] +$$

$$+ \frac{R}{\Phi}\sum_{t=1}^{T}\sqrt{2\alpha\epsilon_t^2\zeta(\kappa, R) + 2\alpha\epsilon_t R} + \frac{R}{\Phi_t}\sum_{t=1}^{T}\|q_{t+1}^* - q_t^*\|$$

$$\leq \frac{R}{\Phi}(d(q_1, q_1^*) - d(q_{T+1}, q_t^*)) + \frac{R}{\Phi}\sum_{t=1}^{T}\sqrt{2\alpha\epsilon_t^2\zeta(\kappa, R) + 2\alpha\epsilon_t R} + \frac{R}{\Phi_t}V_T$$

where the second term can be simplified as:

$$\sum_{t=1}^{T}\sqrt{2\alpha^2\epsilon_t^2 + 2\alpha\epsilon_t R}$$

$$\leq \sum_{t=1}^{T}\sqrt{2\alpha^2\epsilon_t^2} + \sqrt{2\alpha\epsilon_t R}$$

$$\leq \sqrt{2}\alpha\sum_{t=1}^{T}\epsilon_t + \sqrt{2\alpha R}\sum_{t=1}^{T}\sqrt{\epsilon_t}$$

$$\leq \sqrt{2}\alpha E_T + \sqrt{2\alpha R}\sqrt{T\sum_{t=1}^{T}\epsilon_t}$$

$$\leq \sqrt{2}\alpha\zeta(\kappa, R)E_T + \sqrt{2\alpha RTE_T}$$

Then, we finally prove the regret bound as:

$$\mathbb{E}[\mathcal{R}_{\mathcal{P}(\mathcal{W})}(T)] \leq \frac{R^2}{\xi} + \frac{R}{\xi}(\sqrt{2}\alpha\zeta(\kappa, R)E_T + \sqrt{2\alpha RTE_T}) + \frac{R}{\Phi_t}V_T$$

$$\leq \mathcal{O}(\max(1, E_T, V_T))$$

$\square$

Different from the proof of the inexact gradient descent on Euclidean space, we include the trigonometric distance inequality introduced in (Zhang et al., 2016) and give the first dynamic regret bound for the inexact infinitesimal gradient descent methods over $\mathcal{P}_2(\mathcal{W})$. Note the regret bound here is related to a curvature bound $\kappa$, where we set $\kappa$ as a constant since it is not the key point of this paper.

Since the gradient error is denied in $\mathbb{R}^D$, we can follow the same analysis as Section 3.3 to bound the gradient error bound $E_T$, which gives a sublinear error bound. As a result, by setting a sublinear increasing constraint for the variational budget $V_T$, we can make sure $R_{\mathcal{P}_2(\mathcal{W})}(T)$ is increasing sublinear. That means the OPVI methods can converge to the dynamic changing target posterior $p_t$ when $T$ is large enough.

In SVGD, the author didn't consider this gradient error in their algorithm. However, since the gradient error can be viewed as a part of noise added into the updating process, we should not use the whole diffusion noise $\nabla K(x, \cdot)$ in eq. (4). In the experiment, we set the diffusion term as $0.1 \cdot \nabla K(x, \cdot)$ for OPVI. We observe that this trick gives tremendous improvements in performance, especially in some high-dimensional tasks like image classification.

To further find the relationship between $E_T$ and $B_t$ to bound $E_T$, we give some analysis for the gradient error led by the stochastic batch sampling with the sublinear increasing batch size following Theorem 4. Base on section 2.8 in (Lohr, 2021), we have:

$$\mathbb{E}[\|e_t\|^2] = \frac{N_T - B_t}{N_T B_t} \Lambda^2, \tag{11}$$

where $N_T$ is the total number of data samples we have and $\Lambda$ is a bound on the sample variance of the gradients, which is defined by:

$$\frac{1}{N_T - 1} \sum_{i=1}^{N^T} \left\| \nabla c_t^i(\mathbf{w}) - \nabla c_t(\mathbf{w}) \right\|^2 \leq \Lambda^2 \quad \mathbf{w} \in \mathcal{W}$$

To fulfill the requirement of $\mathbb{E}[\|e_t\|^2]$ in eq. (11), we assume $\epsilon_t = \sqrt{\frac{1}{B_t} - \frac{1}{N_T}}$ and the sublinear increasing batch-size as $B_t = \frac{N_T t^\rho}{N_T + t^\rho}$ $\rho > 0$. Then, we can bound $E_T$ as:

$$E_T = \sum_{t=1}^{T} \epsilon_t \leq \sum_{t=1}^{T} (e_t + \eta_t h_t) \leq \sum_{t=1}^{T} \sqrt{\frac{1}{t^q}} + \sum_{t=1}^{T} \eta_t h_t \tag{12}$$

$$\leq \sum_{t=1}^{T} \sqrt{\frac{1}{t^q}} + \sum_{t=1}^{T} \eta_t (w_t - w^*) \leq \frac{2}{2-\rho} T^{1-\frac{\rho}{2}} + R \sum_{t=1}^{T} \|\frac{6}{\pi^2 t^2}\| \tag{13}$$

Here, the noise contains two parts. The first part of the noise comes from the mini-batch sampling. We can see when the batch size $B_t$ is growing sublinear, the gradient error bound $E_T$ is sublinear. Thus if the variational budget $V_T$ is constrained to be sublinear, the regret bound is proved to be sublinear. Note that in the regret analysis, we set a static stepsize $\alpha$ for convenience. The algorithm can also achieve a sublinear regret bound when the stepsize is set to be digressive like $\alpha_t = t^{0.55}$. The second part of the noise comes from the wrongly added prior term. We can observe that the shrinking factor for the prior term is necessary for the convergence of the noise leads by the gradually accumulated prior term error. Next, we illustrate why a static batch size fills to achieve a sublinear regret.

**Remark:** Set the batch size to be static as $B$. The agent update $w_t$ for over $T$ rounds and use a total of $N_T$ data samples, where $N_T$ can be calculated by $N_T = \sum_{t=1}^{T} B = BT$. Following a similar setting in eq. (12), we bound the gradient error over $t \in [1, T]$ as:

$$E_T = \sum_{t=1}^{T} \epsilon_t = \sum_{t=1}^{T} \sqrt{\frac{1}{B} - \frac{1}{N_T}} \leq \sum_{t=1}^{T} \sqrt{\frac{1}{B}(1 - \frac{1}{T})} \leq \mathcal{O}(T)$$

which gives a linear increasing gradient error bound $E_T \leq \mathcal{O}(T)$. That makes it impossible to give a sublinear regret bound, which is necessary to ensure the algorithm can finally converge to the optimal solutions.

## 5 Experiments

In this section, we test the performance of the proposed OPVI algorithm, and compare it with two famous Bayesian sampling methods, the LD (Welling & Teh, 2011) and SVGD (Liu & Wang, 2016). We run these

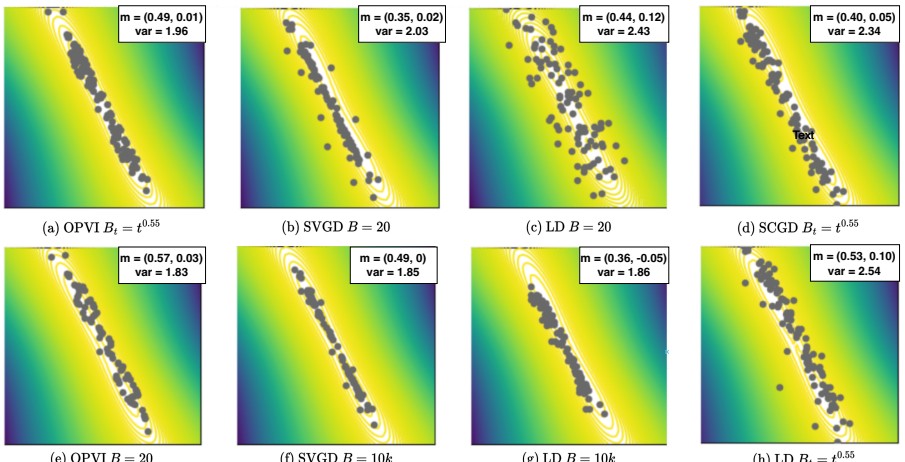

Figure 2: Synthetic experiments for different methods. All methods run 500 rounds. Except for the full batch methods (which use much more data samples), other methods use the same number of data samples. The quantitative mean and variance with respect to the distribution parameters in eq. (14) are shown on the figures.

Table 1: Results on a BNN classification task on the Kin8nm dataset, averaged over 20 tries.

| Methods | Avg. RMSE | Avg. LL | Time |
|---|---|---|---|
| OPVI $B_t = t^{0.55}$ | $.127 \pm .008$ | $.653 \pm .060$ | 2.4 |
| OPVI $B = 20$ | $.145 \pm .003$ | $.516 \pm .021$ | 2.4 |
| SVGD $B = 20$ | $.144 \pm .003$ | $.525 \pm .019$ | 2.4 |
| SVGD $B = 10k$ | $.112 \pm .002$ | $.783 \pm .017$ | 5.8 |
| LD $B = 20$ | $.159 \pm .004$ | $.425 \pm .024$ | 1.7 |
| LD $B = 10k$ | $.143 \pm .002$ | $.527 \pm .015$ | 5.2 |

methods with three types of batch settings, mini-batch with increasing batch size, mini-batch with static batch size, and full batch. To make the comparison fair, we set a Fixed Iterations and Total Data Samples (FITDS) policy for experiments under the mini-batch setting, which means we set the total number of data samples $N_T$ and the total number of time slots $T$ to be same for each experiment.

Except for the full-batch methods, all algorithms follow the FITDS policy. For a dataset of nearly 10k data samples, we run all methods for 500 rounds and set $B = 20$ for the static batch size methods and $B_t = t^{0.55}$ for the increasing batch size methods to keep $N_T$ same. We choose $B_t = t^{0.55}$ as the schedule for the increasing batch-size method since it fulfills the requirement of sub-linear regret in Theorem 8 and test to be the best setting for our method. For full batch methods, we use all 10k data samples in each round to show the best possible results. All experiments are run under the same setting (unless otherwise stated), codes for these experiments are available at `https://github.com/AnonymousSubmission100/OPVI`.

## 5.1 Synthetic Experiments

The synthetic experiments follow the setting in (Welling & Teh, 2011) that conduct a simple example with two parameters, based on the mixture Gaussian distribution:

$$(\theta_1, \theta_2) \sim \mathcal{N}\left((0,0), \text{diag}\left(\sigma_1^2, \sigma_2^2\right)\right)$$
$$x_i \sim 0.5 \cdot \mathcal{N}\left(\theta_1, \sigma_x^2\right) + 0.5 \cdot \mathcal{N}\left(\theta_1 + \theta_2, \sigma_x^2\right), \tag{14}$$

where $\sigma_1^2 = 10$, $\sigma_2^2 = 1$ and $\sigma_x = 2$. Here, we draw approximately 10,000 data samples from the above distribution with $\theta_1 = 0$ and $\theta_2 = 1$. Except for the full-batch methods, all algorithms follow the FITDS

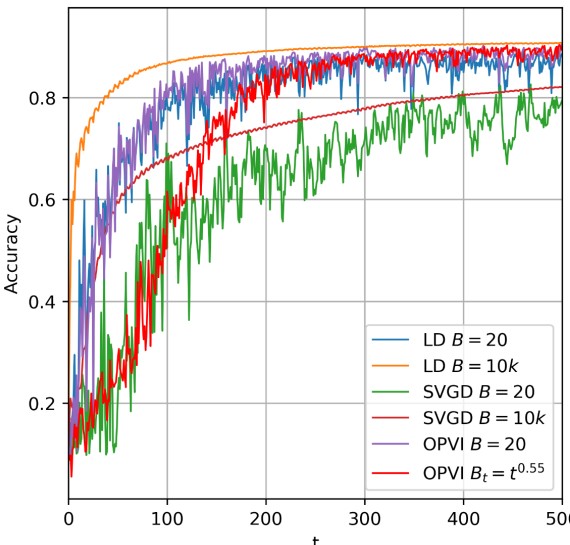

Figure 3: Learning curve of the image classification task. We can observe that the proposed OPVI method outperforms most mini-batch Bayesian sampling methods at the end of training steps

policy. Fig. 2 shows the results for the OPVI, SVGD, and LD with 100 particles, where the true posteriors are shown as contour and the inference results are represented by the particles.

As we can observe from the result, the proposed increasing batch size OPVI gives a better result than the static batch size OPVI, which is caused by the use of increasing batch size as a variance reduction method. Compared with previous SVGD and LD, the OPVI method shows much better performance for tracking the posterior. That should be led by the influence of the gradient noise on the noise injection process of the LD method since we use a smaller diffusion term to offset the gradient error. In the last two figures, we can see the performance of OPVI is approaching or even better than the full batch methods. Also, we can observe from panels (d) and (h) in Fig. 2 that our proposed OVPI method outperforms both the SVGD and LD methods as the batch size increases. This indicates that our innovative design of the prior and the kernel shrinking schedule enhances the performance of the ParVI method in an online setup.

## 5.2 Bayesian Neural Network (BNN) Experiment

In this subsection, we further compare our work with SVGD and LD on some Bayesian Neural Networks (BNN) tasks. We follow the experiment setting in (Liu & Tao, 2015), which uses a single hidden layer BNN with 50 hidden units. We use a Gamma(1, 0.1) function in the prior distribution, Kin8nm as the dataset and divide the dataset randomly 90% for training and 10% for testing. For all methods, we set the number of particles to 20.

All ParVI methods use the same stepsize, except for LD, which uses a smaller but best possible stepsize. We test the Root Mean Squared Error (RMSE) and the test Log-Likelihood (LL). The experiment results are shown in Table. 1. The OPVI algorithm can achieve an **11.8% and 20.1% improvement** compared with SVGD and LD with the same total number of data $N_T$ and the same total time slots $T$ respectively. This result is even comparable to the full batch SVGD algorithm. Note that the running time for OPVI is the same as the SVGD algorithm, which is less than half of the full batch methods.

## 5.3 Image classification Task

Finally, we conduct experiments to test the performance of the proposed algorithm on a high-dimensional image classification problem. The dataset we used is the MNIST dataset, which contains 60,000 training cases and 10,000 test cases. We consider a two-layer BNN model with 100 hidden variables, with a sigmoid

input layer and a softmax output layer. All experiments are using 20 particles. The comparison result is shown in Fig. 3. As we can see from the figure, except for the full batch LD algorithm, the OPVI algorithm with an increasing batch size achieves the best result. However, the full batch LD method uses much more time (30 times) and data samples (500 times), and the result is similar. We can observe that the noise of the increasing batch size OPVI is decreasing with $t$ increase, which verifies our analysis for the gradient error. An interesting thing is that SVGD shows poor performance in this high-dimensional task, which may lead by an incorrect approximation for the diffusion term with limited particle numbers. Instead, we improve the diffusion term in OPVI, which solves this problem.

## 6 Conclusion

In this paper, we consider the OPVI algorithm as a possible sampling method for the intractable posterior under the online setting. The proposed algorithm is the first algorithm to think about the online optimization algorithm from perspective of bayesian sampling and give the theoretical proof to understand the dynamics from the perspective of Wasserstein gradient flow. To reduce the variance, we include an increasing batch size scheme and analyze the influence of the choice of batch size on the performance of the algorithm. Furthermore, we develop a detailed analysis by understanding the algorithm as a Wasserstein gradient flow. Experiments show the proposed algorithm outperforms other naive online particle-based VI and online MCMC methods.

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

# A Proof of Theorem 4

The main idea of this proof follows (Bedi et al., 2018, Theorem 2). We start by providing a Lemma that gives the relationship between the distance $d(w_{t+1}, w_t^*)$ and the quantity $c_t(w_t) + \eta_t c_0(w_t) - c_t(w_t^*) - \eta_t c_0(w_t^*)$. Then, we use this Lemma to prove the Theorem 4.

**Lemma 9.** *Under Assumptions 1 - 3, given a sequence of optimal solutions $\{w_t^*\}$, gradient error $\mathbb{E}[e_t] \leq \epsilon_t$ and the updating policy eq. (3), the online MAP algorithm adheres to the following inequality:*

$$
\begin{aligned}
\mathbb{E}[\|w_{t+1} - w_t^*\|] &\overset{(a)}{\leq} \mathbb{E}[\|w_t - w_t^*\|] \\
&\quad - \frac{\xi}{R}\mathbb{E}[(c_t(w_t) + \eta_t c_0(w_t) - c_t(w_t^*) - \eta_t c_0(w_t^*))] \\
&\quad + \sqrt{2\alpha^2 \epsilon_t^2 + 2\alpha \epsilon_t R},
\end{aligned}
\tag{15}
$$

*where $\xi := 2\alpha - 4L\alpha^2$.*

**Proof.** We start by proving a fact with Assumption 2. For any $w \in \mathcal{W}$, by the smoothness and convexity of $c_t(w) + \eta_t c_0(w)$, in (Zhou, 2018, Lemma 4) we have:

$$
c_t(w) + \eta_t c_0(w) - c_t(w_t) - \eta_t c_0(w_t) \leq \langle \nabla c_t(w_t) + \eta_t \nabla c_0(w_t), w - w_t \rangle + \frac{L}{2}\|w - w_t\|^2
$$

Here, we set a specific value for $w$ as $w = w_t' := w_t - \frac{1}{L}(\nabla c_t(w_t) + \eta_t \nabla c_0(w_t))$ in the above inequality and get:

$$
c_t(w_t') + \eta_t c_0(w_t') - c_t(w_t) - \eta_t c_0(w_t) \leq -\frac{1}{2L}\|\nabla c_t(w_t) + \eta_t \nabla c_0(w_t)\|^2.
$$

On the other hand, by the convexity of $c_t(w)$ and $c_0(w)$ and the vanishing gradient assumption $\nabla c_t(w_t^*) + \eta_t \nabla c_0(w_t^*) = 0$ in Assumption 3, we have:

$$
c_t(w_t') + \eta_t c_0(w_t') \geq c_t(w_t^*) + \eta_t c_0(w_t^*) + (\nabla c_t(w_t^*) + \eta_t \nabla c_0(w_t^*))^\top (w_t' - w_t^*) = c_t(w_t^*) + \eta_t c_0(w_t^*),
$$

which leads to the following inequality of interest:

$$
c_t(w_t^*) + \eta_t c_0(w_t^*) - c_t(w_t) - \eta_t c_0(w_t) \leq c_t(w_t') + \eta_t c_0(w_t') - c_t(w_t) - \eta_t c_0(w_t) \leq -\frac{1}{2L}\|\nabla c_t(w_t) + \eta_t \nabla c_0(w_t)\|^2,
$$

which is equivalent to:

$$
\|\nabla c_t(w_t) + \eta_t \nabla c_0(w_t)\|^2 \leq 2L(c_t(w_t) + \eta_t c_0(w_t) - c_t(w_t^*) - \eta_t c_0(w_t^*)).
\tag{16}
$$

Then, we bound the left-hand side of Lemma 9 by evolving the updating policy eq. (6) in $\|w_{t+1} - w_t^*\|^2$, then:

$$
\|w_{t+1} - w_t^*\|^2 \leq \|\mathcal{P}(w_t - \alpha(\nabla c_t(w_t) + \eta_t \nabla c_0(w_t))) - w_t^*\|^2 \tag{17}
$$

$$
= \|w_t - \alpha(\nabla c_t(w_t) + \eta_t \nabla c_0(w_t)) - w_t^*\|^2 \tag{18}
$$

$$
\overset{(a)}{=} \|w_t - w_t^*\|^2 - 2\alpha(\nabla c_t(w_t) + \eta_t \nabla c_0(w_t))^\top (w_t - w_t^*) + \alpha^2\|\nabla c_t(w_t) + \eta_t \nabla c_0(w_t)\|^2 \tag{19}
$$

$$
\overset{(b)}{\leq} \|w_t - w_t^*\|^2 - 2\alpha(c_t(w_t) + \eta_t c_0(w_t) - c_t(w_t^*) - \eta_t c_0(w_t^*)) \tag{20}
$$

$$
+ 2\alpha^2 L(c_t(w_t) + \eta_t c_0(w_t) - c_t(w_t^*) - \eta_t c_0(w_t^*)) \tag{21}
$$

$$
\tag{22}
$$

where (a) can be obtained by expanding the squared term, (b) is following the convexity property that $f(b) - f(a) \geq \nabla c(a)^\top (b - a)$ and eq. (16).

Finally, we take expectation for $\{w_t^*\}$ as $\mathbb{E}$ and take a root on both sides of the above equation and give:

$$\mathbb{E}[\|w_{t+1} - w_t^*\|] \leq \mathbb{E}[\|w_t - w_t^*\|] - \frac{\xi}{R}\mathbb{E}[(c_t(w_t) + \eta_t c_0(w_t) - c_t(w_t^*) - \eta_t c_0(w_t^*))],$$

where $\xi := 2\alpha - 4L\alpha^2$, the inequality follows $\|w_t - w_t^*\| \leq R$ and the fact $\sqrt{a^2 - b + c^2} \leq a - \frac{b}{2a} + c$ proved as following:

$$\begin{aligned}
\sqrt{a^2 - b} &\leq \sqrt{a^2(1 - \frac{b}{2a^2})^2} \\
&\leq a(1 - \frac{b}{2a^2}) \\
&= a - \frac{b}{2a},
\end{aligned}$$

where $a, b$ are all positive and $a^2 > b$. $\qquad\qquad\square$

Note that if we take a summation over $t \in [1, T]$ on the second term on the right side of eq. (15), we can get the regret $\mathcal{R}(T)$. However, we can divide the left side of eq. (15) into two parts with triangle inequality for a tighter bound, which give the proof for Theorem 4 as follows.

First, we start with using triangle inequality on a quantity $\mathbb{E}[\|w_{t+1} - w_{t+1}^*\|]$, which gives:

$$\begin{aligned}
\mathbb{E}[\|w_{t+1} - w_t^*\|] &\leq \mathbb{E}[\|w_{t+1} - w_t^*\|] + \mathbb{E}[\|w_{t+1}^* - w_t^*\|] \\
&\overset{(a)}{\leq} \mathbb{E}[\|w_t - w_t^*\|] - \frac{\xi}{R}\mathbb{E}[(c_t(w_t) + \eta_t c_0(w_t) - c_t(w_t^*) - \eta_t c_0(w_t^*))] + \|w_{t+1}^* - w_t^*\|
\end{aligned}$$

Rearranging the above inequality and take summation for $t \in [1, T]$, by the definition of the dynamic regret in eq. (2) we have:

$$\begin{aligned}
\mathbb{E}[\mathcal{R}_{\mathcal{W}}(T)] &= \sum_{t=1}^{T}\mathbb{E}[(c_t(w_t) + \eta_t c_0(w_t) - c_t(w_t^*) - \eta_t c_0(w_t^*))] \\
&\overset{(a)}{\leq} \frac{R}{\xi}(\sum_{t=1}^{T}(\mathbb{E}[\|w_t - w_t^*\|] - \mathbb{E}[\|w_{t+1} - w_{t+1}^*\|]) + \sum_{t=1}^{T}\|w_{t+1}^* - w_t^*\|) \\
&\overset{(b)}{\leq} \frac{R}{\xi}(\|w_1 - w_1^*\| - \|w_{T+1} - w_{T+1}^*\| + \sum_{t=1}^{T}\|w_{t+1}^* - w_t^*\|) \\
&\leq \frac{R^2}{\xi} + \frac{RV_T}{\xi}
\end{aligned}$$

where (a) is obtained by using Lemma 9 and (b) follows Assumption 1 and the definition of variational budget $V_T$.

## B   Proof of Lemma 7

*Proof.* We start from a fact proved in Lemma 6 of (Zhang & Sra, 2016), which gives an inequality for a geodesic triangle with curvature bounded by $\kappa$, where the length of sides for the triangle is $a$, $b$, $c$ and $A$ is the angle between sides $b$ and $c$, then:

$$a^2 \leq \frac{\sqrt{|\kappa|c}}{\tanh(\sqrt{|\kappa|c})}b^2 + c^2 - 2bc\cos(A),$$

In our work, we map our problem on a triangle, where the vertices of this triangle is set to be three status of the decisions in our problem, the current step decision $q_t$, the next step decision $q_{t+1}$ and the optimal

solution in current step $q_t^*$. Denote $d(a, b)$ to be the Wasserstein distance between two distribution $a$ and $b$ over $\mathcal{P}_2(\mathcal{W})$. As a result, the three sides of the triangle should be $a = d(q_{t+1}, q_t^*)$, $b = d(q_t, q_{t+1})$ and $c = d(q_t, q_t^*)$. Base on the updating rule, we have $d(q_t, q_{t+1}) = \alpha\|\nabla c_t(q_t) + e_t + \nabla c_t^0(q_t)\|$ when $\alpha$ is small enough and $d(q_t, q_{t+1})d(q_t, q_t^*)\cos(\angle q_{t+1}q_tq_t^*) = \langle -\alpha(\nabla c_t(q_t) + e_t + \nabla c_t^0(q_t)), \mathrm{Exp}_{q_t}^{-1}(q_t^*)\rangle$. Taking all sides into the triangle inequality, we have:

$$d(q_{t+1}, q_t^*)^2 \overset{(a)}{=} \zeta(\kappa, d(q_t, q_t^*))d(q_t, q_{t+1})^2 + d(q_t, q_t^*)^2 - 2\langle\alpha(\nabla c_t(q_t) + e_t + \eta_t\nabla c_t^0(q_t) + \eta_t h_t, \mathrm{Exp}_{q_t}(q_t^*)\rangle$$
$$\leq \zeta(\kappa, d(q_t, q_t^*))(\alpha(\nabla c_t(q_t) + e_t + \eta_t h_t + \eta_t\nabla c_t^0(q_t))^2 + d(q_t, q_t^*)^2 - 2\langle\alpha(\nabla c_t(q_t) + e_t + \eta_t h_t + \eta_t\nabla c_t^0(q_t), \mathrm{Exp}_{q_t}(q_t^*$$
$$\leq d(q_t, q_t^*)^2 + \zeta(\kappa, d(q_t, q_t^*))(\alpha^2(\nabla c_t(q_t) + \eta_t\nabla c_t^0(q_t))^2 + \alpha^2(e_t + \eta_t h_t)^2 + 2\alpha^2(e_t + \eta_t h_t)(\nabla c_t(q_t) + \eta_t\nabla c_t^0(q_t)))$$
$$- 2\alpha\langle(\nabla c_t(q_t) + \eta_t\nabla c_t^0(q_t)), \mathrm{Exp}_{q_t}(q_t^*)\rangle - 2\alpha\langle(e_t + \eta_t h_t), \mathrm{Exp}_{q_t}(q_t^*)\rangle$$
$$\overset{(b)}{\leq} d(q_t, q_t^*)^2 + \zeta(\kappa, R)(\alpha^2(\nabla c_t(q_t) + \eta_t\nabla c_t^0(q_t))^2 + \alpha^2(e_t + \eta_t h_t)^2 + 2\alpha^2(e_t + \eta_t h_t)(\nabla c_t(q_t) + \eta_t\nabla c_t^0(q_t)))$$
$$- 2\alpha\langle(\nabla c_t(q_t) + \eta_t\nabla c_t^0(q_t)), \mathrm{Exp}_{q_t}(q_t^*)\rangle - 2\alpha\langle(e_t + \eta_t h_t), \mathrm{Exp}_{q_t}(q_t^*)\rangle$$
$$\overset{(c)}{\leq} d(q_t, q_t^*)^2 + \zeta(\kappa, R)(2L\alpha^2(c_t(q_t) + \eta_t c_t^0(q_t) - c_t(q_t^*) - \eta_t c_t^0(q_t^*)) + \alpha^2(e_t + \eta_t h_t)^2 + 2\alpha^2(e_t + \eta_t h_t)(\nabla c_t(q_t) + \eta_t\nabla$$
$$- 2\alpha(c_t(q_t) + c_t^0(q_t) - c_t(q_t^*) - c_t^0(q_t^*)) - 2\alpha\langle(e_t + \eta_t h_t), \mathrm{Exp}_{q_t}(q_t^*)\rangle$$
$$= d(q_t, q_t^*)^2 + 2L\alpha^2\zeta(\kappa, R)(c_t(q_t) + c_t^0(q_t) - c_t(q_t^*) - c_t^0(q_t^*)) + \zeta(\kappa, R)\alpha^2(e_t + \eta_t h_t)^2$$
$$- 2\alpha(c_t(q_t) + c_t^0(q_t) - c_t(q_t^*) - c_t^0(q_t^*)) + 2\alpha^2(e_t + \eta_t h_t)\zeta(\kappa, R)(\nabla c_t(q_t) + \nabla c_t^0(q_t)) - 2\alpha\langle(e_t + \eta_t h_t), \mathrm{Exp}_{q_t}(q_t^*)\rangle$$

where (a) follows (Zhang & Sra, 2016, Lemma 6), $\zeta(\kappa, d(q_t, q_t^*)) = \frac{\sqrt{|\kappa|}d(q_t, q_t^*)}{\tanh(\sqrt{|\kappa|}d(q_t, q_t^*))}$, (b) follows the assumption 5, (c) follows the fact proved in eq. (16) and the convexity of $c_t(q_t) + c_t^0(q_t)$ Then, we bound the last two terms in the above inequality with the expectation on the sequence of $\{e_t\}$, which is denoted by $\mathbb{E}_{e_t}$ and get::

$$\mathbb{E}[2\alpha^2(e_t + \eta_t h_t)\zeta(\kappa, R)(\nabla c_t(q_t) + \nabla c_t^0(q_t)) - 2\langle(e_t + \eta_t h_t), \mathrm{Exp}_{q_t}(q_t^*)\rangle]$$
$$\leq 2\alpha^2\zeta(\kappa, R)\mathbb{E}[e_t + \eta_t h_t](\nabla c_t(q_t) + \nabla c_t^0(q_t)) + 2\alpha\mathbb{E}[e_t + \eta_t h_t]R$$
$$\overset{(a)}{\leq} \alpha^2\zeta(\kappa, R)(\epsilon_t^2 + \mathbb{E}[(\nabla c_t(q_t) + \nabla c_t^0(q_t))]^2) + 2\alpha\epsilon_t R$$
$$\leq \alpha^2\zeta(\kappa, R)(\epsilon_t^2 + c_t(q_t) + c_t^0(q_t) - c_t(q_t^*) - c_t^0(q_t^*)) + 2\alpha\epsilon_t R,$$

where (a) is obtained by the fact $2ab \leq a^2 + b^2$. Taking the above inequality back gives:

$$\mathbb{E}_{e_t}[d(q_{t+1}, q_t^*)]^2 \leq d(q_t, q_t^*)^2 + 2L\alpha^2\zeta(\kappa, R)(c_t(q_t) + c_t^0(q_t) - c_t(q_t^*) - c_t^0(q_t^*)) + \zeta(\kappa, R)\alpha^2\epsilon_t^2$$
$$- 2\alpha(c_t(q_t) + c_t^0(q_t) - c_t(q_t^*) - c_t^0(q_t^*))$$
$$+ \alpha^2\zeta(\kappa, R)(\epsilon_t^2 + 2Lc_t(q_t) + c_t^0(q_t) - c_t(q_t^*) - c_t^0(q_t^*)) + 2\alpha\epsilon_t R$$
$$= d(q_t, q_t^*)^2 - \Phi(c_t(q_t) + c_t^0(q_t) - c_t(q_t^*) - c_t^0(q_t^*)) + 2\alpha\epsilon_t^2\zeta(\kappa, R) + 2\alpha\epsilon_t R,$$

where $\Phi = 2\alpha - 3L\alpha^2\zeta(\kappa, R)$.

Finally, using the fact $\sqrt{a^2 - b + c^2} \leq a - \frac{b}{2a} + c$ and full expectation $\mathbb{E}$, we finish the proof:

$$\mathbb{E}[d(q_{t+1}, q_t^*)] \leq \mathbb{E}[[\mathbb{E}_{e_t}[d(q_{t+1}, q_t^*)^2]]^{1/2}]$$
$$\leq \mathbb{E}[d(q_t, q_t^*)] - \frac{\Phi}{R}\mathbb{E}[(c_t(q_t) + c_t^0(q_t) - c_t(q_t^*) - c_t^0(q_t^*))] + \sqrt{2\alpha\epsilon_t^2\zeta(\kappa, R) + 2\alpha\epsilon_t R}$$

$\square$

