# OpenReview forum: "Particle-based Online Bayesian Sampling"
_TMLR — Rejected by TMLR_

### Review · Reviewer_5WYv · 2024-04-03

**Summary Of Contributions:**

In this paper, the authors propose a new algorithm for online learning. Their goal is to sample from the posterior model of a Bayesian model using an online learning procedure.

The paper has three main contributions:

1. An upper bound of the dynamic regret

2. A new weight for the prior decay (proto 1/t^2)

3. Empirical studies of the proposed method

**Audience:**

Yes

**Claims And Evidence:**

No

**Requested Changes:**

I see two paths forward to this paper:

1 If we assume the proposed algorithm adds little to the existing algorithms. The authors could concentrate on the theoretical aspects of the paper. Extend the theoretical analysis to most of the ParVI methods mentioned in the literature review, and they should also analyze how tight these bounds are. If there are ways in which we can use them to improve convergence.

2 If the authors believe the proposed algorithm is significantly different, they should make an algorithmic comparison with LD and SVGD and point out the differences between the proposed algorithm and the two competitors. Also, they should compare all the algorithms on the same conditions: fix mini-batch, growing mini-bath, full-batch.

**Strengths And Weaknesses:**

The main strength of the paper lies in the theoretical results (Theorem 4 and Theorem 8). However, these are not contextualized with past theoretical results. Unfortunately, I am not an expert on these bounds and cannot judge their relevance.

The paper, in other ways, is hard to judge. The authors emphasize that this is the first online method of its kind, and they mentioned that ParVI is used in batch and mini-batch settings. However, there is no difference between mini-batch and online learning. It will be good that the authors describe where mini-batch and online differ.

Also, in the experimental setting, the proposed online algorithm with 20 samples performs as the two ParVI methods with 20-sample mini-batches. The authors try to indicate that their proposed algorithm is better than the mini-batch version of LD and SVGD when, for the first two cases, their algorithm performs as SVGD and, in the last case, as LD when they use 20-sample minibatches. Also, I would not expect SVGD to underperform in this experiment (why is the case?). I would expect LD and SVGD would perform as the proposed method with a growing mini-batch.

The authors also point out that they handle the prior better, but besides proposing a decaying function, it is not detailed what they do to handle the prior better. If this proposal handles the prior better, it would be good to know why and what the other methods are doing wrong with their handling of the prior.

The authors also point out that they handle the prior better, but besides proposing a decaying function, it is not detailed what they do to handle the prior better. If this proposal handles the prior better, it would be good to know why and what the other methods are doing wrong with their handling of the prior.

---

> ### Author Response · Authors · 2024-05-11
> **Response to the weakness and comments**
>
> Thanks for the valuable suggestions provided by the review. Here are our responses to the weakness and comments. We are sorry for the late reply, we misunderstood the rebuttal rules and waited for the third review at first.
>
> # Weakness
>
> > The authors emphasize that this is the first online method of its kind, and they mentioned that ParVI is used in batch and mini-batch settings. However, there is no difference between mini-batch and online learning. It will be good that the authors describe where mini-batch and online differ.
>
> This is a great point. The main difference is how the training data is provided. In the mini-batch SGD used in standard offline training, the whole training dataset is provided at the very beginning and it does not change in the training process. In our online setting, the training data set is provided sequentially, the training data set keeps increasing in the training process. This significant difference in the problem setup will cause entirely different theoretical guarantees, which is defined by the so-called regret to measure the convergence in the long run (t:1->T)
>
> > I would not expect SVGD to underperform in this experiment (why is the case?). I would expect LD and SVGD would perform as the proposed method with a growing mini-batch.
>
> Thank you for the comments. Our proposed method is not simply an adaptation of the SVGD method with streaming datasets and increasing batch sizes. As discussed in Section 3.1 and the paragraph above eq. (11), we also introduce a novel configuration of specific kernel and prior shrinking factors, tailored specifically for online learning scenarios.
>
> Further experiments are added in Fig. 2 of the revision to demonstrate that our method continues to outperform SVGD and LD even with increasing batch size, which shows the importance of our newly designed shrinking factors. Moreover, as shown in Fig.~2 of the paper, even when employing full-batch training (B=10k) for SVGD and LD, which represents a more noise-free scenario, these methods still underperform compared to our proposed method.
>
> > The authors also point out that they handle the prior better, but besides proposing a decaying function, it is not detailed what they do to handle the prior better. If this proposal handles the prior better, it would be good to know why and what the other methods are doing wrong with their handling of the prior.
>
> The simplest way to understand our decaying setup for the prior term is to consider the difference between the online Bayesian sampling and the offline case. For every step of update in the online case, the prior should change according to the posterior obtained last step to make use of the previously information. Thus, the addition of the original prior term should be shrinking concerning the steps, as more and more information provided by data samples is accumulated.
>
> To show the benefit mathematically, we edited our proof in Lemma 7 and Theorem 8 in the revision, which includes the error caused by the wrong addition of the prior term. We can observe that the reduction of the noise leads by the original prior term, especially in eq. 13 of the revision.
>
>
> # Changes
>
> > . Extend the theoretical analysis to most of the ParVI methods mentioned in the literature review, and they should also analyze how tight these bounds are. If there are ways in which we can use them to improve convergence.
>
>
> We believe the tightness is not a problem since the optimization problem has been widely considered in the traditional online convex optimization field over the Euclidean space. Our main contribution is to extend the proof of online convex optimization to the Bayes case over the Wasserstein space. To enable this proof, we propose a new triangle inequality over the Wasserstein space used specifically for our problem in Lemma 7. Furthermore, we analyze the noise leads by the min-batch sampling and the prior term.
>
> For the tightness of the online optimization problem over the Euclidean space, we can refer to papers like e.g. ‘Tracking Slowly Moving Clairvoyant: Optimal Dynamic Regret of Online Learning with True and Noisy Gradient’, where the lower bound matches the upper bound in their setup.  Given the limited time, we cannot extend the lower bound proof in the revision and we believe it’s not the main focus of our paper.

---

### Review · Reviewer_h1iX · 2024-04-08

**Summary Of Contributions:**

This paper proposes to use online learning techniques to perform Bayesian sampling.
The paper builds on a set of methods known as particle-based variational inference, which approximates the posterior using a set of particles. The proposed method is called Online Particle-based Variational Inference (OPVI), which operates on the Wasserstein space.
They prove results on the dynamic regret of OPVI.
The paper shows the method on synthetic experiments and a BNN experiment.

**Audience:**

Yes

**Claims And Evidence:**

No

**Requested Changes:**

Please see the weaknesses section.

Also, I think the paper should undergo a rewrite to address clarity issues. As a reader, I cannot easily grasp various motivations throughout the paper and overall don't find the paper to be well-written.

**Strengths And Weaknesses:**

Strengths
* Bayesian inference is known to be a hard problem and therefore, the problem studied here is interesting and potentially impactful.

Weaknesses
* What is the motivation for an online version of ParVI? In the introduction of the paper it is stated that “there is no online version of ParVI” and then immediately “we develop OPVI to meet this desideratum”. The paper would be improved if clearer motivation were given at the beginning.

* Regarding Assumption 2 (convexity and Lipschitz smooth cost function); this seems very strong. Can you give some examples of Bayesian sampling where the convexity assumption holds?

* Motivation for OPVI on Wasserstein space? It was unclear to me why OPVI  was chosen. Perhaps this is because I am unfamiliar with Wasserstein gradient flow, but in any case, I think the paper could have properly introduced the motivation and walked the reader through these choices.

* Throughout the paper, writing could be improved. I recommend the authors go through the paper and improve the writing for clarity and readability of the paper. As of now, it does not feel like a coherently written paper.
    * Pg 2: “succedaneum” (use a better word?)
    * The figures were not referenced formally
    * Proofs don’t need to be in the main paper

---

> ### Author Response · Authors · 2024-04-09
> **Response to the weakness and comments (1 out of 2)**
>
> Thank you very much for your review. We recognize that due to its theoretical nature, our paper may require additional background knowledge for full comprehension. To address this, we have uploaded our revision to enrich the manuscript with more background information. Herein, we address your concerns to improve the overall understanding of our work in the upcoming revision.
>
> # Weaknesses
>
> > What is the motivation for an online version of ParVI? In the introduction of the paper it is stated that “there is no online version of ParVI” and then immediately “we develop OPVI to meet this desideratum”. The paper would be improved if clearer motivation were given at the beginning.
>
> Thank you for the comment. We further explained the motivation for studying the online Bayesian sampling method in revision. Generally, this research is motivated by many application scenarios where the data is collected sequentially and one has to update the uncertainty-aware learning model on the fly. Specific examples include: (1) Bayesian matrix/tensor completion with streaming measurement data (e.g., in MRI), (2) Bayesian sparse or low-rank learning of surrogate models with screaming scientific simulation data (e.g., in high-dim uncertainty quantification), (3) Bayesian on-device learning in an uncertain environment with streaming sensor data. We have added these motivations to section 1 of the revision and referred to related publications about the details.
>
> > Regarding Assumption 2 (convexity and Lipschitz smooth cost function); this seems very strong. Can you give some examples of Bayesian sampling where the convexity assumption holds?
>
> The convexity assumption is commonly employed in analyzing ParVI methods, particularly when conceptualizing the process theoretically as an optimization problem of the KL divergence. Due to the enduring challenges associated with solving non-convex optimization problems and the lack of a comprehensive theoretical foundation in the Wasserstein space, nearly all papers in this field opt for an analysis based on the convex case. Thus, this assumption is made even though the posterior in the sampling problem may exhibit both convex and non-convex characteristics. We provide references to several papers that analyze ParVI under the convexity assumption.
>
> - Sharrock, Louis, and Christopher Nemeth. "Coin Sampling: Gradient-Based Bayesian Inference without Learning Rates." International Conference on Machine Learning. PMLR, 2023.
> - Liu, Chang, et al. "Understanding and accelerating particle-based variational inference." International Conference on Machine Learning. PMLR, 2019.
> - Sharrock, Louis, Lester Mackey, and Christopher Nemeth. "Learning Rate Free Bayesian Inference in Constrained Domains." Advances in Neural Information Processing Systems 36 (2024).
>
> Additionally, the smoothness assumption is almost universally adopted in this field to facilitate the study of optimization problems. Conversely, for discontinuous functions that exhibit infinite gaps in function value at certain points, achieving a convergence rate is fundamentally unattainable.

---

> ### Author Response · Authors · 2024-04-09
> **Response to the weakness and comments (2 out of 2)**
>
> > Motivation for OPVI on Wasserstein space? It was unclear to me why OPVI was chosen. Perhaps this is because I am unfamiliar with Wasserstein gradient flow, but in any case, I think the paper could have properly introduced the motivation and walked the reader through these choices.
>
> Thank you for your suggestion. We opted to investigate the OPVI method in Wasserstein space because our optimization objective diverges from the loss function used in standard optimization (measured with two vectors in real space, as exemplified by the MAP example in Section 3). Instead, the OPVI method's optimization objective is the KL-divergence, which measures the difference between two distributions (see Section 4). This distinction means we cannot directly frame OPVI as a conventional optimization problem in Euclidean space. Building on the insights from the paper by Chang Liu, we are able to model the ParVI problem as a gradient optimization problem over Wasserstein space. This approach facilitates theoretical analysis from an optimization perspective. Consequently, we chose to focus on online optimization in the Wasserstein space, leading to the derivation of the first regret bound for online optimization in this context. We also acknowledge the importance of adding some descriptions for this part and we have added them at the end of the introduction in the revision.
>
>
> > Throughout the paper, writing could be improved. I recommend the authors go through the paper and improve the writing for clarity and readability of the paper. As of now, it does not feel like a coherently written paper.
>
> Thank you for your suggestion.  We have further polished the paper and updated the revision. Generally, we updated the following staff:
>  - We correct the general typos in the revision
> - We correct the confusing term in the equations in eq. (1), Theorem 4, eq. (7) and (8) and the corresponding proof in the appendix.
> - We add further clarification about the motivation, discussion about theorem 4, and experiments.
>
> We highlight all the corrections in the revision in blue color.
>
> Regarding the term "succedaneum" mentioned on Page 2, we will opt for a more straightforward term such as "replacement" to enhance clarity. We include certain proofs directly in the main text because the theoretical guarantees represent a core contribution to our paper. Important proofs are thus highlighted, with thorough analysis provided for these critical sections. As for figure references, we further polished the caption of the figures.
>
> Feel free to follow-up if there are further questions, thanks a lot!
>
> Best,
>
> Authors

---

### Review · Reviewer_5jPQ · 2024-05-05

**Summary Of Contributions:**

The authors present an Online Particle-based Variational Inference (OPVI) method that improves upon existing ParVI methods in online settings through an increasing batch-size scheme and refined prior-term settings. This work also provides a theoretical analysis of the OPVI method, which is the first theoretical analysis for ParVI from the perspective of dynamic regret by leveraging the Riemannian structure of the Wasserstein space.

**Audience:**

Yes

**Claims And Evidence:**

Yes

**Requested Changes:**

Clarity could be significantly enhanced, particularly regarding Equation (1), which appears confusing due to the fixed set of observations seemingly independent of the time index 't'. Specifying the true data distribution model at the outset of Section 3.1, including how the N_T samples relate to the parameterized distribution 'w' and its prior, would clarify this issue. Currently, the online nature of Equation (1) isn't evident, as the samples don't represent streaming data. Additionally, decomposing the optimization problem into T separate instances for different 'eta_t' values raises questions about the necessity of solving T MAP solutions, each for a different 'eta_t', instead of under a single 'w' parameter.

The undefined gradient error bound E_T in Theorem 4 requires clarification, particularly since it doesn't contribute to the regret bound results. Furthermore, it's essential to explicitly state that 'alpha' represents the step size in Theorem 1, and the theorem itself needs restating for improved clarity. Moreover, additional discussions or interpretations of the results in Theorem 4 are necessary.

In Section 4.3, by definition of KL divergence, it should be ${D_{KL}}_p(q_t)$ = $\mathbb{E}[\log q_t] - \mathbb{E}[\log p]$, and the same correction applies to Equation (8).

Regarding particle updates, only a single iteration is performed at each time step. It could be worth exploring whether increasing the number of iterations would enhance theoretical results or empirical performance.

Regarding Figure 2, clarity is needed on whether the plotted particles represent those obtained at the last time step T. Furthermore, quantitative results comparing the performance of different methods should be included alongside Figure 2, as the methods appear similar in the plot, making it challenging to discern which is superior.

Typos:
In Section 3.2, “where ΠW is the projection back to the convex set W. projection, need an additional inequality in proof "
In Section 4.2, “vector field approximation, need to be verified”
On page 10, "$N^T$" should be "$N_T$."

**Strengths And Weaknesses:**

Strengths: A new online particle-based VI method for posterior sampling.

Weakness: However, clarity could be significantly enhanced. This paper is not clearly written and lacks descriptions and clarification. Also, the significance of the problem studied is not sufficiently justified.

---

> ### Author Response · Authors · 2024-05-11
> **Response to the weakness and comments**
>
> ###  We wish to extend our heartfelt gratitude for the substantial time and effort you have dedicated to reviewing our manuscript. We have addressed your concerns and uploaded the revision, all the changes made to the manuscript have been marked in blue color.
>
> >  Weakness: However, clarity could be significantly enhanced. This paper is not clearly written and lacks descriptions and clarification. Also, the significance of the problem studied is not sufficiently justified.
>
> We have improved the clarity of the paper, please see the following clarification and the updated revision for details. We also stress the significance of the online Bayes problem in the revision, which can be summarized as follows.
>
> Significance of the online Bayes problem:
>
> Generally, this research is motivated by many application scenarios where the data is collected sequentially and one has to update the uncertainty-aware learning model on the fly. Specific examples include: (1) Bayesian matrix/tensor completion with streaming measurement data (e.g., in MRI), (2) Bayesian sparse or low-rank learning of surrogate models with screaming scientific simulation data (e.g., in high-dim uncertainty quantification), (3) Bayesian on-device learning in an uncertain environment with streaming sensor data. We have added these motivations to section 1 of the revision and referred to related publications about the details.
>
> ### Clarification: Thanks for the suggestions, we have corrected all typos. Please refer to the revision. Here are some key points of our update.
> > Equation (1), which appears confusing due to the fixed set of observations seemingly independent of the time index 't'. Specifying the true data distribution model at the outset of Section 3.1, including how the N_T samples relate to the parameterized distribution 'w' and its prior, would clarify this issue. Currently, the online nature of Equation (1) isn't evident, as the samples don't represent streaming data
>
> For the representation of data samples in eq. (1), we replace the definition of dataset with a set of steaming data batches, which makes the online nature of the dataset clear.
>
> > Additionally, decomposing the optimization problem into T separate instances for different 'eta_t' values raises questions about the necessity of solving T MAP solutions, each for a different 'eta_t', instead of under a single 'w' parameter.
>
> Necessary for solving T MAP solutions: We don’t think it’s necessary to solve T MAP solutions. When we update a single set of parameters and gradually involve the prior term, it can enable a stable and accurate update toward the posterior for the Bayes system by including the information obtained from the previous data. Even though the true posterior is changing in the online setting, the change is smooth, which makes continuous updates of a single parameter necessary.
>
> > The undefined gradient error bound E_T in Theorem 4 requires clarification
>
> Error bound E_T: We are sorry for the typo, we didn’t consider the E_T term in the online MAP demo. We have deleted the description of the E_T term from the theorem in the revision.
>
> > Moreover, additional discussions or interpretations of the results in Theorem 4 are necessary.
>
> Discussion for Theorem 4: We have added the discussion about Theorem 4 in the revision. We discuss the non-linearity of the regret bound concerning the time step t.
>
> > In Section 4.3, by definition of KL divergence
>
> Definition of KL-Divergence: We have corrected the definition of KL-Divergence in section 4.3 and eq. (8) in the uploaded revision. Our original idea is to represent the minus KL divergence. Sorry for the confusion.
>
> > Regarding particle updates, only a single iteration is performed at each time step. It could be worth exploring whether increasing the number of iterations would enhance theoretical results or empirical performance.
>
> Regarding Performance Improvement: Increasing the number of iterations can indeed enhance performance. However, given that the distribution changes slightly at each step (L-smoothness), a single update step strikes a better balance between efficiency and performance.
>
> >  Regarding Figure 2, clarity is needed on whether the plotted particles represent those obtained at the last time step T
>
> Feedback on Particle Statistics: Thank you for the suggestion. The particles are obtained at the last time step T. We have provided the mean and variance in Fig.2 regarding the distribution given in eq. (13). From these quantitative results, we can observe that the proposed OPVI method can provide better performance for the online Bayesian sampling case.
>
> >Typos: In Section 3.2, “where ΠW is the projection back to the convex set W. projection, need an additional inequality in proof " In Section 4.2, “vector field approximation, need to be verified” On page 10, "$N^T$" should be "$N_T$."
>
> All these typos have been fixed.
>
> Thanks,
>
> Authors

---

### Author Response · Authors · 2024-05-14
**Rebuttal and Revision Uploaded**

Dear Reviewers and Editors,

Thank you very much for reviewing our paper. We have completed the rebuttal and updated the revision accordingly. We apologize for the delayed response and appreciate your patience throughout this process. We are grateful for all the feedback received. Please feel free to reach out if you have any further questions.

Best regards,

The Authors

---

### Decision · Action_Editor_6DKL · 2024-06-17

**Recommendation:** Reject

**Comment:**

There were no concerns about correctness, but all reviewers seemed to be confused about the paper's claims, specifically the motivation, how it compares to related work, and what is its significance. There were also concerns about novelty being limited. None of the reviewers supported accepting the paper, with 2 voting "Leaning reject" and 1 "Reject".

The authors might want to resubmit the paper after a major revision, which would require them to frame the work in a way that would appeal to some audience at TMLR, perhaps by finding an interesting application or a better motivation for the setting.

**Audience:**

There appear to be issues with the audience for this submission. Even before the reviewing process started, it was very hard to find appropriate reviewers for the paper as many declined due to the work being outside of their area. Even the reviewers who eventually agreed to review the submission didn't have a perfect match with the paper either, see, for instance, the comment by Reviewer 5WYv, which states that they are not an expert on the topic. Beyond that, Reviewer 5jPQ expressed concern that the significance of the problem considered in this work is not well explained. Reviewer h1iX also noticed that the motivation behind the paper was unclear. Reviewer 5WYv in their final recommendation stated that some of the key aspects of this work are "uninteresting for the problem they propose to solve."

Based on these concerns, I recommend rejecting the paper in accordance with TMLR's criterion 2, which states that a paper should be accepted when there is a large enough audience at TMLR and which I believe is not satisfied here.

**Claims And Evidence:**

The paper seems to have correct claims with sufficient evidence. Reviewer 5WYv stated that the work is not contextualized with past theoretical results properly, and Reviewer h1iX expressed having a similar opinion in their final decision. Reviewers mention difficulties with understanding the setting, the motivation behind the setting and the assumptions. While one reviewer wrote that the claims are correct and the theoretical results are novel, another stated the novelty to be limited. Overall, all reviews seem to suggest that the paper, as Reviewer 5WYv wrote, "is hard to judge", and one reviewer responded "No" to the question of claims being supported by evidence in their final recommendation.

**Resubmission Of Major Revision:**

The authors may consider submitting a major revision at a later time.